# Infectious Bronchitis Virus Activates the Aryl Hydrocarbon Receptor During In Vitro Infection

**DOI:** 10.3390/vetsci12100932

**Published:** 2025-09-24

**Authors:** Mingjing Zhang, Zhichao Cai, Hongliu An, Rong He, Songbai Zhang, Shouguo Fang

**Affiliations:** College of Agriculture, Yangtze University, Jingzhou 434025, China; echozh@hotmail.com (M.Z.); caizhichao110@126.com (Z.C.); 1984625825@163.com (H.A.); coco-hey@outlook.com (R.H.)

**Keywords:** infectious bronchitis virus, aryl hydrocarbon receptor, host therapeutic target

## Abstract

Avian coronaviruses, such as infectious bronchitis virus (IBV), pose significant threats to poultry health and the economic viability of poultry farms by evading host immune defenses. Similarly to their human counterparts, IBVs may exploit the aryl hydrocarbon receptor (AhR) to suppress antiviral immune responses. Through the use of cell-based models and pharmacological agents targeting AhR, we found that IBV activates AhR to promote viral replication and exacerbate inflammatory responses. Pharmacological inhibition of AhR significantly reduced viral replication, whereas AhR activation showed no significant effect. Genetic knockdown of AhR expression further confirmed its role in facilitating IBV infection. These findings suggest that AhR-targeting therapeutics may offer a promising strategy for protecting poultry from IBV infection. Given the similarities in immune evasion strategies between avian and human coronaviruses, targeting AhR may not only help mitigate economic losses in poultry farming but also inform the development of antiviral interventions for human coronaviruses.

## 1. Introduction

Coronaviruses (CoVs), which belong to the family *Coronaviridae* within the order *Nidovirales*, are enveloped, single-stranded RNA viruses that cause multisystem pathologies in the respiratory, gastrointestinal, and nervous systems of both animal and human hosts [1,2]. The subfamily *Coronavirinae* is classified into four genera—*Alphacoronavirus*, *Betacoronavirus*, *Gammacoronavirus*, and *Deltacoronavirus* [3].

Avian infectious bronchitis virus (IBV), a member of the *Gammacoronavirus* genus, causes a highly contagious disease known as avian infectious bronchitis [4]. The virus primarily replicates in the respiratory epithelium but also exhibits variable tropism for the kidneys, intestines, and oviduct tissues, thereby imposing significant economic challenges on the global poultry industry [5]. IBV was the first coronavirus to have its complete 27.6 kb genome sequenced, which is characterized by a standard 5′-capped structure and a 3′ polyadenylated tail [6,7]. Its genomic organization enables the production of six subgenomic mRNAs through a nested transcription mechanism. These mRNAs encode 15 nonstructural proteins (nsp2-nsp16), four structural proteins, and four accessory proteins [8,9].

The aryl hydrocarbon receptor (AhR), a ligand-activated basic helix-loop-helix/Per-Arnt-Sim (bHLH-PAS) transcription factor, demonstrates a broad ligand-binding capacity for structurally diverse xenobiotics, tryptophan metabolites, and indoles derived from the microbiota [10,11]. As a molecular sensor, AhR coordinates a wide range of biological functions, including cellular proliferation, metabolic reprogramming, and, most notably, plays a pivotal role in immunomodulation through interactions with both innate and adaptive immune pathways [12,13]. In its inactive state, AhR exists in the cytoplasm as a multiprotein complex composed of the chaperone heat shock protein 90 (HSP90), the co-chaperone p23, and Hepatitis B virus X-associated protein 2 (XAP2) [14]. Upon ligand binding, the AhR translocates to the nucleus, where it forms a heterodimer with the aryl hydrocarbon receptor nuclear translocator (ARNT) [11]. This results in a transcriptionally active complex that binds to xenobiotic response elements (XREs) [15]. This AhR-ARNT complex mediates the transcriptional activation of phase I and phase II metabolic enzymes through XRE-driven promoter binding, particularly inducing members of the cytochrome P450 family (CYP1A1, CYP1B1, CYP2A1), which are essential for xenobiotic detoxification [16,17].

Emerging evidence identifies the aryl hydrocarbon receptor (AhR) as a key regulator of virus–host interactions [18]. Its signaling axis exhibits proviral activity during flavivirus infections, such as Zika virus (ZIKV) and dengue virus (DENV), primarily through the suppression of type I interferon (IFN-I)-mediated antiviral defenses [19,20]. Murine hepatitis virus (MHV) infection activates AhR via IDO1-independent pathways, which in turn modulates viral replication kinetics and the host’s interferon-stimulated gene (ISG) response [21]. Pharmacological inhibition of AhR using CH223191 has been shown to significantly reduce viral titers of SARS-CoV-2 and human coronavirus 229E (HCoV-229E) in vitro, coinciding with the activation of the AhR pathway during infection [22]. Mechanistically, AhR activation induced by SARS-CoV-2 enhances viral fitness by suppressing IFN-I responses and increasing the expression of the ACE2 receptor.

Collectively, these findings identify AhR activation as an evolutionarily conserved immune evasion mechanism utilized by coronaviruses to suppress IFN-I responses and improve viral replicative fitness [23]. Targeted inhibition of AhR emerges as a promising host-directed, broad-spectrum therapeutic approach, deserving further preclinical evaluation across zoonotic and pandemic coronavirus strains [22].

Although direct evidence linking AhR signaling to IBV pathogenesis is currently lacking, the phylogenetic conservation of immune-modulatory mechanisms within the *Coronaviridae* family supports the likelihood of its involvement. Both pharmacological inhibition of AhR using selective antagonists and genetic silencing via shRNA-mediated knockdown significantly suppressed IBV replication efficiency, as demonstrated by plaque assays and viral RNA quantification. These results highlight AhR as a novel therapeutic target for the development of IBV-specific antiviral strategies and underscore the need for further investigation into its spatiotemporal regulation during avian coronavirus infection.

## 2. Materials and Methods

### 2.1. Cell Culture and Transfections

The human non-small cell lung carcinoma cell line H1299 was purchased from the Cell Bank of the Committee on Type Culture Collection of Chinese Academy of Sciences (CSTR: 19375.09.3101HUMTCHu160), monkey kidney epithelial cells were sourced from ATCC(CCL-81). Both cell lines were propagated in serum-supplemented media (RPMI-1640 for H1299; DMEM for Vero) (RPMI-1640, DMEM, Gibco, Grand Island, NY, USA) containing 10% fetal bovine serum (FBS, Gibco, Grand Island, NY, USA), with routine incubation at 37 °C under 5% CO_2_ [24].

H1299 cells were seeded in 6-well plates and cultured under standard conditions until reaching 70–80% confluency. For lipofection, 2 µg of shRNA plasmid (Invitrogen, Waltham, MA, USA) and 5 µL Lipofectamine 3000 (Invitrogen, Waltham, MA, USA) were separately diluted in 250 µL Opti-MEM (Gibco, Grand Island, NY, USA), incubated for 5 min at room temperature, then combined and incubated for 15 min to form transfection complexes. The complexes were gently added dropwise to the cell culture medium. After 6 h of transfection, the medium was replaced with fresh complete DMEM.

### 2.2. Viral Strains

Wild-type recombinant infectious bronchitis virus (rIBV), a Vero cell-adapted strain (IBV Beaudette P65; GenBank: DQ001339.1), was generated via reverse genetics and produced noncoding RNA (ncRNA) [25]. The mutant strain rIBV-C27107G, derived from the parental IBV-p65 backbone through site-directed mutagenesis, lacks ncRNA production [26].

Virus stocks were generated by infecting cells at a multiplicity of infection (MOI) of 0.1 in serum-free DMEM for 24–36 h until 70–80% cytopathic effect (CPE) was observed [25]. Following three freeze–thaw cycles, supernatants were clarified by centrifugation, aliquoted, and stored at −80 °C.

### 2.3. Small Molecule Compounds

The AhR antagonist CH223191 (1-Methyl-N-[2-methyl-4-[2-(2-methylphenyl) diazinyl] phenyl-1H-pyrazole-5-carboxamide) (Sigma-Aldrich, St. Louis, MO, USA) was dissolved in DMSO and tested at 1.25, 2.5, 5, 10, 20, 40, and 80 µM [27].

The AhR agonist L-kynurenine (2S)-2-Amino-4-(2-aminophenyl)-4-oxobutanoic acid) (Sigma-Aldrich, St. Louis, MO, USA) was reconstituted in DMSO and applied at 2.5, 5, 10, 20, 40, 80, 160, and 320 µM [28].

All photosensitive compounds were handled under light-restricted conditions.

### 2.4. Cell Viability Assays

H1299 and Vero cells were treated for 24 h with CH223191 (1.25 µM to 80 µM), L-kynurenine (2.5 µM to 320 µM), or equivalent DMSO concentrations (vehicle control). All experimental procedures were performed in accordance with the CCK-8 Cell Counting Kit (Vazyme, Nanjing, China). The absorbance at 450 nm was measured using an enzyme immunoassay analyzer. Cell viability % was determined by relativizing treatment absorbance results using Mock absorbance values. Cellular morphology and viability were assessed via light microscopy at 200× magnification.

### 2.5. Pharmacological Treatments and Viral Infections

Cells pretreated with CH223191 (5 µM to 80 µM) or L-kynurenine (2.5 µM to 320 µM) for 24 h were infected with rIBV or rIBV-C27107G (MOI of 0.5). Supernatants and cells were harvested at 48 h post-infection for plaque assays, qRT-PCR, or Western blot.

### 2.6. Viral Nucleic Acid Quantification

Total RNA isolated with TRIzol (Invitrogen, Waltham, MA, USA) was reverse-transcribed using PrimeScript™ RT Master Mix (Takara Bio Inc., Shiga, Japan). Quantitative real-time PCR (qRT-PCR) was performed with SYBR^®^ Green (Takara Bio Inc., Shiga, Japan)under cycling conditions: 95 °C for 3 min; 40 cycles of 95 °C for 30 s, primer-specific annealing for 1 min, and 72 °C for 1 min. Relative gene expression was calculated via the 2^−ΔΔCt^ method normalized to *GAPDH*. The primer sequences used for qRT-PCR are listed in Table 1.

### 2.7. Western Blotting

Cells lysed in RIPA buffer (containing 1% PMSF) (Thermo Fisher Scientific, Waltham, MA, USA) were resolved on 10% SDS-PAGE gels and transferred to PVDF membranes (Stratagene, San Diego, CA, USA). After blocking with 5% skim milk, membranes were probed with anti-AhR (Abclonal, Wuhan, China) (1:1000), anti-β-actin (Abclonal, Wuhan, China) (1:5000), or anti-IBV-N (1:8000) antibodies, followed by horseradish peroxidase (HRP)-conjugated secondary antibodies (Abclonal, Wuhan, China) (1:5000). Protein bands were visualized using an enhanced chemiluminescence (ECL) detection kit (Thermo Fisher Scientific, Waltham, MA, USA) and quantified via ImageJ (National Institutes of Health, Bethesda, MD, USA). β-actin was used as the control.

### 2.8. Plaque Assay

Viral titers were determined by infecting Vero cell monolayers with serially diluted supernatants for 2 h. Cells were overlaid with 0.4% agarose-DMEM and incubated for 36–72 h. Plaques were stained with 0.2% crystal violet after formaldehyde fixation. Titers (PFU/mL) were calculated from triplicate experiments.

### 2.9. Statistical Analysis

Quantitative data analysis was conducted in GraphPad Prism v8.0 employing ordinary least squares regression models (GraphPad Software, San Diego, CA, USA). Data were expressed as Mean  ±  SEM. One-Way ANOVA with Dunnett’s post hoc test by was performed to evaluate statistical significance.

## 3. Results

### 3.1. IBV Infection Activates AhR Signaling

Transcriptomic profiling of H1299 cells infected with either wild-type IBV (rIBV, ncRNA-producing) or the mutant strain rIBV-C27107G (ncRNA-deficient) identified 1357 differentially expressed genes, including 740 upregulated and 617 downregulated genes [29]. Subsequent qRT-PCR validation confirmed that IBV infection, regardless of its capacity to produce ncRNA, induces activation of the AhR pathway. This activation is characterized by increased expression of canonical downstream effectors (*CYP1A1*, *CYP1B1*) and pro-inflammatory cytokines (*IL-6*, *IL-8*, *IL-12β*), leading to the induction of antiviral responses in H1299 cells. Importantly, the ncRNA encoded by IBV negatively modulates AhR signaling, thereby attenuating the expression of *IL-6*, *IL-8*, and *IL-12β*. This suppression of the host’s antiviral immune response facilitates viral replication [30].

Previous studies have suggested that AhR functions as a proviral host factor that various viruses exploit to counteract IFN-I-mediated immune defenses [18]. The spatiotemporal regulation of AhR activation involves complex, multilevel mechanisms, highlighting the need for further investigation into its molecular basis and potential as a therapeutic target. While additional research is needed to fully characterize AhR’s role in avian coronaviruses, this study establishes its functional involvement during in vitro IBV infection and highlights critical regulatory interactions for future investigation.

### 3.2. Pharmacological Targeting of AhR Modulates Viral Replication

To elucidate the regulatory role of the AhR in IBV infection, we assessed the effects of pharmacological modulation of AhR using its canonical antagonist CH223191 and agonist L-kynurenine (kynurenine) on in vitro infections with two IBV variants: the wild-type rIBV and the ncRNA-deficient mutant rIBV-C27107G. Parallel experiments were carried out in H1299 and Vero cells; the latter lacks the ability to produce type I interferon (IFN-I), thereby allowing the distinction of IFN-I-dependent mechanisms involved in AhR-mediated host–virus interactions. Cellular viability under increasing concentrations of CH223191 (1.25–80 µM) and L-kynurenine (2.5–320 µM) was assessed using the MTT assay and further confirmed through morphological evaluation by light microscopy (Figure 1). Dose–response curves were constructed based on optical density (OD570) measurements to determine the cytotoxic thresholds of the respective compounds.

CH223191 induced significant cytotoxicity in H1299 cells at 80 µM, accompanied by cell rounding and membrane blebbing, while Vero cells exhibited only a mild reduction in viability without apparent morphological alterations (Figure 1A,C). Kynurenine exhibited no toxicity in either cell line (Figure 1B,D).

To investigate AhR-mediated pharmacological modulation during IBV infection, Vero and H1299 cell cultures were treated with dimethyl sulfoxide (vehicle), CH223191 (2.5 µM, 5 µM, 10 µM, and 20 µM), or kynurenine (5 µM, 10 µM, 20 µM, and 40 µM), administered during pre- and post-infection phases. Recombinant IBV strains (rIBV and rIBV-C27107G) were inoculated at MOI of 0.5, with viral titers determined by PFU quantification following 48 h supernatant collection (Figure 2).

Pharmacological inhibition of AhR by CH223191 demonstrated significant antiviral effects against both IBV strains. In Vero and H1299 cells treated with CH223191, viral titers of rIBV and rIBV-C27107G decreased in a dose-dependent manner compared to DMSO controls (Figure 2A,B). In contrast, pretreatment with kynurenine did not alter viral replication at any of the tested concentrations, as plaque counts were comparable to those of the vehicle-treated groups (Figure 2C,D). These findings collectively indicate that pharmacological inhibition of AhR significantly suppresses IBV replication in vitro, suggesting that AhR is a key host factor involved in coronavirus propagation.

### 3.3. AhR Modulation Impacts IBV-N Protein Expression

To investigate the mechanistic role of AhR in IBV replication, we quantified AhR expression and viral nucleocapsid (N) protein levels in parallel using pharmacologically modulated H1299 (IFN-competent) and Vero (IFN-deficient) cell systems. The IBV-encoded N protein, a phylogenetically conserved structural component, plays a central role in viral genome packaging through the formation of ribonucleoprotein (RNP) complexes and actively participates in replication and transcription processes [31].

Vero and H1299 cells were pretreated with vehicle (DMSO), CH223191 (10 µM), or kynurenine (40 µM) prior to infection with rIBV or rIBV-C27107G. Cellular lysates collected 48 h post-infection were analyzed by Western blot for AhR and IBV-N protein levels. Protein band intensities were quantified using ImageJ software (Version 1.53q; NIH, USA).

As shown in Figure 3A, pretreatment with CH223191 significantly reduced AhR protein expression in H1299 cells infected with either rIBV or rIBV-C27107G, compared to the DMSO-treated controls. Notably, the suppression of AhR was more evident in cells infected with rIBV than in those infected with rIBV-C27107G (Figure 3B), indicating that the IBV-encoded ncRNA may partially attenuate the downregulation of AhR. In contrast, kynurenine treatment did not significantly affect AhR protein levels under any of the tested conditions.

Consistent with these observations, Western blot analysis confirmed a dose-dependent inhibition of IBV nucleocapsid (N) protein synthesis in CH223191-treated cells. Both H1299 and Vero cells showed significantly reduced IBV-N expression at 10 µM CH223191 (Figure 3C,D), correlating with the decreased viral titers in Figure 2. In contrast, kynurenine had no effect on IBV-N levels despite its AhR-activating capacity (Figure 3C,D).

These findings demonstrate that IBV infection activates AhR through both viral noncoding RNA (ncRNA)-dependent and -independent pathways. The distinct AhR expression patterns in wild-type versus mutant IBV infections suggest ncRNA modulates AhR signaling dynamics. Furthermore, AhR perturbation caused stronger N protein suppression in Vero cells, highlighting IFN-independent regulatory mechanisms.

### 3.4. AhR Suppression Reduces IBV Viral Levels

To investigate whether pharmacological inhibition of AhR by CH223191 affects the IBV RNA expression, Vero and H1299 cells were pretreated with CH223191 (10 µM) or kynurenine (40 µM) for 48 h, followed by infection with rIBV or rIBV-C27107G. Total RNA extracted post-infection was analyzed by qRT-PCR to measure RNA levels of *AhR*, *CYP1A1*, and the viral nucleocapsid gene (Figure 4). In both cell lines infected with either viral strain, CH223191 significantly reduced *AhR* mRNA levels compared to the DMSO controls, whereas kynurenine markedly increased *AhR* transcripts (Figure 4A,B). *CYP1A1* mRNA levels in H1299 cells showed similar trends: downregulated by CH223191 and upregulated by kynurenine (Figure 4C,D). Notably, CH223191 suppressed *IBV-N* RNA accumulation in both cell types, while kynurenine enhanced viral RNA levels (Figure 4E,F).

Our data collectively demonstrate that pharmacological inhibition of AhR signaling by CH223191 significantly impairs the replication of both wild-type IBV (rIBV) and its noncoding RNA-deficient mutant (rIBV-C27107G) in vitro. The inverse correlation between AhR activity and viral RNA levels supports AhR as a key host factor promoting IBV infection.

### 3.5. AhR Knockdown Inhibits IBV-N Expression

To investigate the role of endogenous AhR in IBV replication, we constructed two short hairpin RNAs (shRNAs) targeting human AhR mRNA (shAhR-1403: 5′-GCTTTGTTTGCGATAGCTACT-3′; shAhR-1934: 5′-GCTACCACATCCACTCTAAGC-3′). These shRNAs were cloned into a pLKO.1-GFP lentiviral vector using BamHI and EcoRI restriction sites (New England Biolabs, Inc., Ipswich, MA, USA). In transfected cells, AhR mRNA and protein levels were significantly reduced (Figure 5), with shAhR-1403 showing highest silencing efficiency and selected for subsequent studies.

Next, H1299 cells transfected with shAhR-1403 were infected with rIBV and rIBV-C27107G at a MOI of 0.5. Western blot analysis revealed baseline IBV-N protein upregulation in unmodified cells, but AhR knockdown suppressed IBV-N expression (Figure 6). This inverse correlation between AhR activity and viral protein synthesis suggests AhR facilitates IBV replication, providing mechanistic evidence for its role in the viral life cycle.

## 4. Discussion

The genomic plasticity of IBV—characterized by high rates of recombination and mutation—drives the persistent emergence of novel genotypes and serotypes with limited cross-protective immunity, posing a major challenge to vaccine development [32]. This rapid evolution underscores the critical need for novel antiviral approaches targeting evolutionarily conserved host–pathogen interaction mechanisms. There is a growing body of evidence that one of the essential molecular players modulating immune response to viral infection might be the AhR.

Our findings demonstrate that IBV infection activates AhR, while similar to conserved AhR activation patterns observed in other coronavirus infections, including SARS-CoV-1 and HCoV-229E. SARS-CoV-2 employs AhR to downregulate IFN-I responses and upregulate ACE2 receptor expression, whereas pharmacological inhibition of AhR like CH223191 effectively suppresses in vitro and in vivo replication of SARS-CoV-2 variants [18,22]. Similarly, in feline coronavirus (FCoV), the AhR antagonist CH223191 provoked a reduction in FCoV replication and in the levels of viral nucleocapsid protein [33]. This mechanism is consistent with our findings in IBV. The CH223191 (10 µM) treatment of IBV-infected (rIBV and rIBV-C27107G) H1299 and Vero cells reduced viral titers by 56%, 79.5% (rIBV), and 75%, 83% (rIBV-C27107G), in compared to DMSO controls (Figure 2A,B), accompanied by marked decreases in IBV-N protein expression (Figure 3C,D) and viral RNA levels (Figure 4E,F). On the other hand, the kynurenine (40 µM) treatment showed no significant effects on viral titers and IBV-N expression (Figure 2C,D and Figure 3C,D). The consistent antiviral effect observed across both cell lines suggests AhR facilitates IBV replication through IFN-I-independent mechanisms.

Preliminary data demonstrate distinct AhR expression patterns between wild-type and mutant IBV infections, suggesting that noncoding RNA (ncRNA) modulates AhR signaling dynamics. Specifically, IBV-encoded ncRNA partially suppresses virus-induced AhR upregulation, thereby inhibiting the expression of pro-inflammatory cytokines (IL-6, IL-8, and IL-12β) [30]. This suppression attenuates the host antiviral response and facilitates viral replication. Furthermore, IBV likely exploits ncRNA-mediated regulation of host gene expression to evade IFN-dependent immune defenses, representing an evolutionary strategy to enhance viral fitness.

AhR activation serves as a common strategy for most viruses to evade antiviral immunity and enhance viral replication. Although the underlying mechanisms of viral activation or suppression of AhR vary, their functional objectives are unequivocal: to either escape host immune responses or promote self-replication and survival. In SARS-CoV-2 infection, the virus upregulates indoleamine 2,3-dioxygenase 1 (IDO1) through promoted expression of IFN-β and IFN-γ, thereby activating the AhR signaling pathway. Reciprocally, AhR activation further enhances viral replication and exacerbates pulmonary lesions [22]. ZIKV activates the AhR pathway to suppress IFN-I expression and reduce antiviral capacity. Concurrently, AhR activation inhibits the NF-κB signaling pathway and restricts PML protein-driven intrinsic immunity, collectively facilitating ZIKV replication in vivo [19]. These mechanistically distinct yet functionally convergent strategies underscore AhR’s pivotal role as a critical host–pathogen interface across viral taxa.

Although the precise interplay between the AhR pathway and canonical antiviral signaling networks (e.g., IFN-I, NF-κB) during IBV infection remains elusive, viral-host interaction paradigms suggest potential crosstalk in modulating apoptosis, cellular homeostasis, and immune responses. Emerging evidence implicates AhR activity in pulmonary pathogenesis and reproductive/renal dysfunction, with implications for poultry health management. As a master regulator of immunomodulation and oxidative stress, AhR orchestrates vital physiological processes through ligand-specific activation (3,3′-diindolylmethane, DIM), demonstrating dual roles in avian development and inflammatory regulation [34]. Mono(2-ethylhexyl) phthalate (MEHP)-mediated AhR activation impairs ovarian antral follicles by suppressing estrogen synthesis and disrupting steroidogenic pathways, effects reversible by CH223191 antagonism [35]. AhR activation enhances nuclear translocation and expression of nuclear factor erythroid 2-related factor 2 (NRF2), reduces reactive oxygen species (ROS) accumulation, and inhibits ferroptosis, providing a potential therapeutic target for the treatment of acute kidney injury (AKI) [36].

Preliminary findings demonstrate that pharmacological blockade using the AhR antagonist CH223191 or endogenous AhR suppression via recombinant plasmid transfection, significantly inhibits IBV replication during infection. These interventions provide mechanistic evidence supporting AhR’s regulatory role in IBV pathogenesis. However, comprehensive elucidation of AhR-mediated mechanisms underlying IBV pathogenesis necessitates the establishment of AhR-specific knockout models coupled with integrated metabolomic-proteomic profiling, enabling multi-tiered dissection of its immunoregulatory networks through orthogonal analytical modalities. Crucially, comparative studies with non-coronaviral avian pathogens—notably Newcastle disease virus (NDV) and avian influenza viruses (AIV)—are imperative to evaluate AhR’s therapeutic universality. Such investigations could catalyze the development of broad-spectrum antiviral agents with dual utility in poultry biosafety management and zoonotic pandemic prevention.

Furthermore, evaluating AhR antagonism in vivo represents a priority research direction. For viruses utilizing AhR activation to enhance antiviral responses (e.g., influenza A virus, IVA), AhR blockade compromises immune defenses and exacerbates infection progression [37]. Conversely, in viruses leveraging AhR signaling to promote replication (e.g., SARS-CoV-2, ZIKV), AhR inhibition enhances antiviral immunity and restricts viral propagation, as evidenced by reduced viral titers and attenuated pathological lesions following antagonist treatment [19,22].

Investigating viral-specific modulation of the AhR pathway will advance our understanding of virally hijacked pathogenic mechanisms, thereby enabling discovery of precision-targeted therapeutic interventions against viral infections. Importantly, chronic AhR blockade may disrupt immune-metabolic homeostasis, necessitating development of pathogen-tailored AhR modulators. Strategic combination of these agents with direct-acting antivirals could achieve dual objectives: suppression of viral replication machinery concurrent with preservation of host immunoregulatory networks. This approach demonstrates significant potential to overcome multi-drug resistance (MDR) barriers in viral therapeutics.

## 5. Conclusions

In conclusion, the combined results of the present study identify AhR as a promising therapeutic target for the development of IBV-specific antiviral strategies, extending its known role in SARS-CoV-2 pathogenesis to avian coronaviruses.

## Figures and Tables

**Figure 1 vetsci-12-00932-f001:**
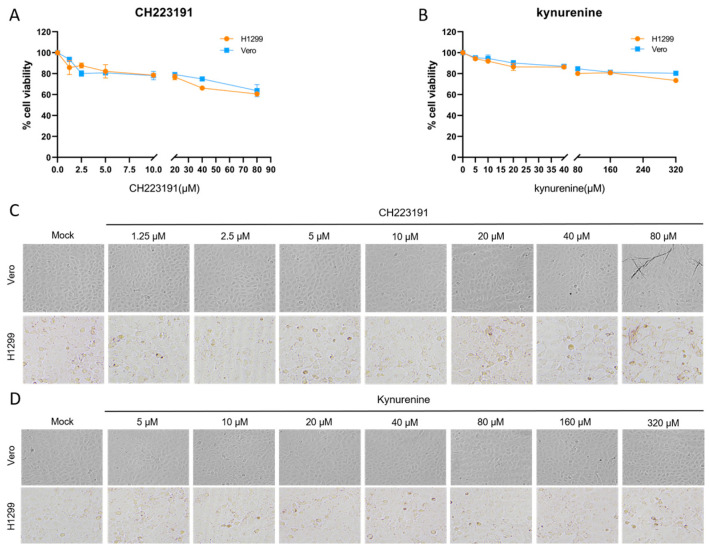
Cytotoxicity profiling of experimental compounds. (**A**,**C**) Cells were treated with gradient concentrations of CH223191 or (**B**,**D**) kynurenine for 48 h. Cellular viability was quantified via MTT assay (**A**,**B**), with mock-treated controls receiving equivalent volumes of vehicle (0.1% DMSO). (**C**,**D**) Representative bright-field micrographs illustrate cytomorphological alterations following 48 h compound exposure, captured under 200× magnification using phase-contrast microscopy.

**Figure 2 vetsci-12-00932-f002:**
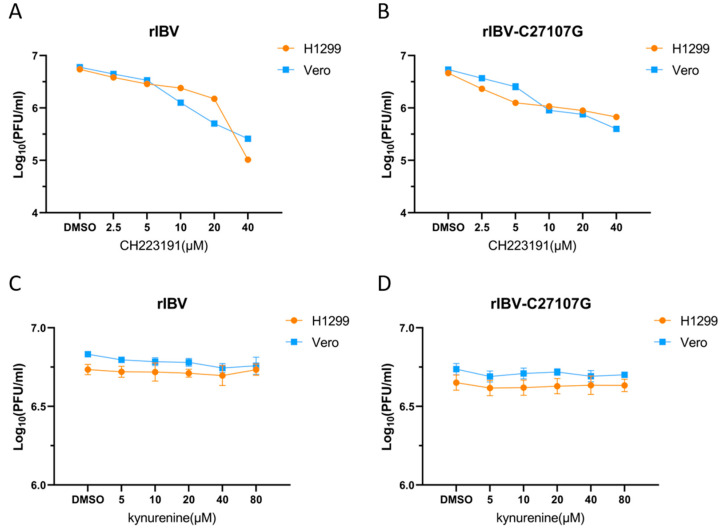
Impact of two compounds on viral yield in IBV-infected cells. Cells were pretreated with CH223191 (**A**,**B**) or kynurenine (**C**,**D**), and subsequently infected with either recombinant IBV (**A**,**C**) or the mutant strain rIBV-C27107G (**B**,**D**). The DMSO-treated groups (0.1% DMSO) were used as experimental controls. Statistical analysis was performed by comparing treatments with vehicle control.

**Figure 3 vetsci-12-00932-f003:**
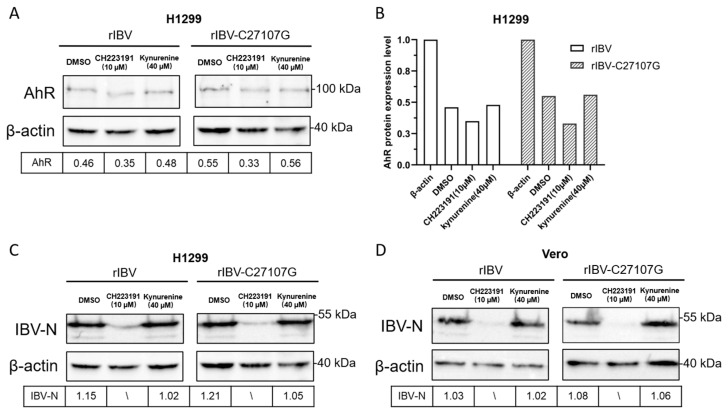
Effects of CH223191 and kynurenine on AhR and IBV-N protein expression levels. H1299 (**A**,**C**) and Vero (**D**) cells were pretreated with CH223191 or kynurenine, respectively, and subsequently infected with either rIBV or the mutant strain rIBV-C27107G (MOI of 0.5) for a duration of 48 h. The protein expression levels of AhR and IBV-N were analyzed using Western blotting. (**B**) Quantitative assessment of AhR protein expression normalized to β-actin levels.

**Figure 4 vetsci-12-00932-f004:**
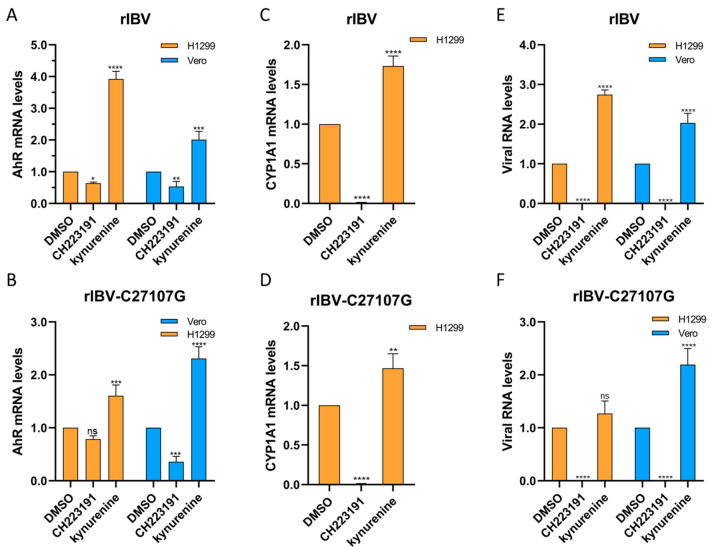
Modulatory effects of CH223191 and kynurenine on expression levels of target genes in virus-infected Vero and H1299 cells. H1299 and Vero cells were pre-treated with either CH223191 or kynurenine, followed by infection with rIBV (**A**,**E**) or rIBV-C27107G (**B**,**F**) for a duration of 48 h. The mRNA expression levels of AhR (**A**,**B**), CYP1A1 (**C**,**D**), and IBV-N (**E**,**F**) were quantified using qRT-PCR. Bar graphs illustrate the relative expression ratios (2^−ΔΔCt^) of the target genes in compound-treated groups normalized to the DMSO control group, with GAPDH used as the endogenous reference gene. Results were plotted as the mean ± SD. *p* < 0.05 was considered statistically significant. ns, *p* > 0.05, non-significant; * *p* < 0.05; ** *p* < 0.01; *** *p* < 0.001; **** *p* < 0.0001.

**Figure 5 vetsci-12-00932-f005:**
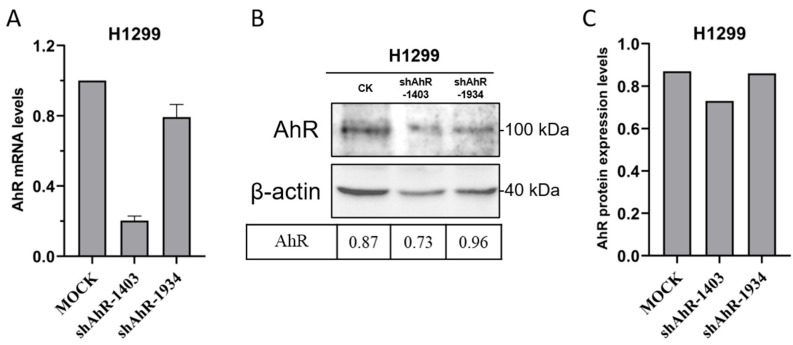
Molecular Validation of AhR Knockdown Efficiency. The silencing efficiency of the shAhR constructs was quantitatively evaluated through qRT-PCR analysis (**A**) at the transcriptional level (**B**,**C**). Quantitative densitometric analysis of AhR protein expression was performed, with normalization to β-actin levels.

**Figure 6 vetsci-12-00932-f006:**
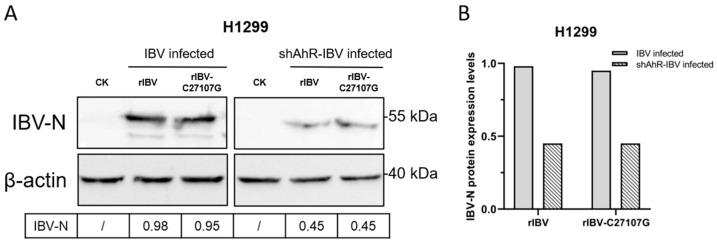
Impact of AhR knockdown on IBV-N protein expression. (**A**) Western blot analysis was performed on H1299 cells transduced with shAhR-1403 and subsequently infected with dual-strain rIBV/rIBV-C27107G (MOI of 0.5) for 48 h. (**B**) Quantitative densitometric analysis of IBV-N protein expression was conducted, with results normalized to β-actin levels.

**Table 1 vetsci-12-00932-t001:** Primer sequences.

Gene	Primer	Sequences (5′-3′)	Position	Accession Number
*AhR*	F	TTGTGTTTCCTAAATCCAACCATT	3807–3830	NM_001621.5
	R	GCAGTTAACAGCAGATTTTTCAC	4154–4132	
*CYP1A1*	F	CTCAGCTCAGTACCTCAGCCAC	71–92	NM_000499.5
	R	CCCCATACTGCTGGCTCATC	328–309	
*IBV-N*	F	TGAAGGTAGCGGTGTTCCTG	26,025–26,044	NC_001451.1
	R	CCACGGTTCAGGGGAATGAA	26,360–26,341	
*GAPDH*	F	GTCAAGGCTGAGAACGGGAA	346–365	NM_002046.7
	R	AGTGATGGCATGGACTGTGG	714–695	

## Data Availability

The original contributions presented in this study are included in the article/Appendix A. Further inquiries can be directed to the corresponding author(s).

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
