# Peer review of "Infectious Bronchitis Virus Activates the Aryl Hydrocarbon Receptor During In Vitro Infection"

_vetsci, 2025, doi:10.3390/vetsci12100932_

Round 1
Reviewer 1 Report
Comments and Suggestions for Authors
Manuscript vetsci-3832876. ¨Infectious Bronchitis Virus Activates the Aryl Hydrocarbon Re-ceptor During In Vitro Infection¨
Manuscript by Zhang et al, describes similarities between signaling pathways upon IBV (avian) and SARS Cov2 infections (humans). Pharmacological approach with substances targeting aryl hydrocarbon receptor (AhR) in a cell-culture based model showed same modulation and thus convergence of both type on coronaviruses. Findings support possible development of broad-spectrum antiviral therapies, considering conserved pathways as targets for counterbalancing the high genetic variability of IBV, and other coronaviruses in mammals.The eventual strategic role that this type of modulation could provide for infection control and disease management in general.
Experimental design was properly followed, and the findings are quite interesting, however it is in need for further refinement before get published. Please see below. Best regards !.
_______________________________________________________________________
Text of the manuscript is very well written in shape and content. Claims are quite well supported by the experimental findings. Methods are adequate to achieve the conclusions. Statistical treatment was properly conducted and supported the conclusions. Particular attention must be paid to the discussion section.
Discussion section is in need to be rewritten. Please address quantitative results and to relate those findings with similar studies in order to compare and properly assess the contribution of these experiments to the global interfering approach.
MAJOR FINDINGS:
-Please discuss for IBV, which has been recognized as an extra-respiratory pathogen, how the AhR pathway could be related with pathogenesis to the reproductive and renal systems.
-Please discuss how much of the findings could be extrapolated at the organism level. Is the AhR pathway contributing somehow to associated lung pathology in chickens ?
-Lanes 119-123. Please describe in detail how accurate the MOI 0.5 could be reached using the IBV recombinant clone.
-Lanes 184-189. Signaling pathways leading to apoptosis have been reported activated through the AhR in other coronaviruses models. Please discuss how apoptosis could ¨useful¨ for the infection purposes in the case of IBV affecting lower respiratory tract ?.
-Lanes 268-269. In addition to IFN-I-driven antiviral immune mechanisms, please discuss findings as related to other antiviral responses such as suppression of NF-κB-based pathways.
Fig 5:
-AhR-specific RNA was noticed to be shutdown by 80% (Fig 5A), and the encoded protein only in 20% (Fig 5C). Are these silencing efficiencies sufficient for the analysis of AhR dependency ?. Please support discussion with references.
Author Response
We thank the reviewer for this valuable suggestion. We have incorporated several relevant points raised by the reviewers into the discussion section. Please see the attachment.

Reviewer 2 Report
Comments and Suggestions for Authors
This manuscript described "Infectious Bronchitis Virus Activates the Aryl Hydrocarbon Receptor During In Vitro Infection". It appeared to be a good topic and some studies seemed interestingly. However, several questions have been raised during the reviewing. I would like to ask for a detailed response before any further consideration can be reached.
Specific questions were as follows:
- For viral nucleic acid quantification, the author should address how much RNA used for cDNA synthesis and how much it is used for qRT-PCR?
- In table 1. Primer sequences, please provide the position of the primer on the nucleotide, accession number, and references.
- If the AhR pathway helps trigger the body's antiviral response but is also used by the virus to its advantage, could blocking AhR actually weaken the immune system and make the infection worse? What does this mean for using AhR as a target for antiviral treatments?
- Since CH223191 showed cytotoxic effects in H1299 cells at higher doses, how can we be sure that the observed reduction in viral replication isn’t partly due to impaired cell viability rather than specific inhibition of AhR? Would using alternative, less toxic AhR inhibitors help clarify this?
- If CH223191 treatment reduced both AhR and viral N protein levels, how do the author distinguish whether the drop in N protein is a direct result of AhR inhibition or just a downstream consequence of reduced overall AhR expression? Would using genetic knockdown (e.g., siRNA) of AhR alongside pharmacological inhibition help clarify the specific mechanism?
- The fact that kynurenine, an AhR agonist, failed to increase N protein levels or enhance viral replication raises the question: is AhR activation alone insufficient to promote IBV replication without additional cofactors? Would testing other AhR agonists or combining kynurenine with IFN pathway inhibitors help clarify this mechanism?
- The observation that kynurenine enhances viral RNA levels suggests a possible proviral role for AhR activation, but could this effect be influenced by broader metabolic changes induced by kynurenine rather than AhR signaling alone? Would metabolic profiling or use of more selective AhR agonists help clarify this?
- While AhR knockdown clearly reduced IBV-N protein levels, was there any assessment of off-target effects or potential cellular stress responses induced by shRNA transfection that could independently affect viral replication? Including a scrambled shRNA control or performing transcriptome analysis could help confirm the specificity of the observed effect.
- Expand on how IBV’s ability to activate AhR independently of IFN signaling represents an evolutionary strategy to enhance viral replication. Please discuss the specific role of the viral ncRNA in modulating AhR signaling dynamics, and how this fine-tuning helps the virus evade immune responses while promoting replication. Comparing this mechanism to other coronaviruses like SARS-CoV-2 could highlight conserved and unique features of coronavirus-host interactions. Expand on AhR as a promising broad-spectrum antiviral target given its role in IBV and other coronaviruses. Address benefits and concerns like cytotoxicity and the need for in vivo studies in avian models. Consider how AhR inhibition could complement current antivirals while noting AhR’s broader immune and metabolic functions.
Author Response

(The authors gave the same response as above.)
